# Sensor Data Fusion in Multi-Sensor Weigh-In-Motion Systems

**DOI:** 10.3390/s20123357

**Published:** 2020-06-13

**Authors:** Janusz Gajda, Ryszard Sroka, Piotr Burnos

**Affiliations:** Department of Measurement and Electronics, AGH University of Science and Technology, Al. A. Mickiewicza 30, 30-059 Cracow, Poland; jgajda@agh.edu.pl (J.G.); burnos@agh.edu.pl (P.B.)

**Keywords:** data fusion, weigh in motion (WIM) systems, multi-sensor WIM, accuracy of WIM systems

## Abstract

In this paper, we present the results of a comparison of two estimators of the gross vehicle weight (GVW) and the static load of individual axles of vehicles. The estimators were used to process measurement data derived from Multi-Sensor Weigh-In-Motion systems (MS-WIM). The term estimator is understood as an algorithm according to which the dynamic axle load measurement results are processed in order to determine the static load. The result obtained is called static load estimate. As a measure of measurement uncertainty, we adopted the standard deviation of the static load estimate. The mean value and the maximum likelihood estimators were compared. Studies were conducted using simulation methods based on synthetic data and experimental data obtained from a WIM system equipped with 16 lines of polymer axle load sensors. We have shown a substantially lower uncertainty of estimates determined using the maximum likelihood estimator. The results obtained have considerable practical significance, particularly during long-term usage of multi-sensor WIM systems.

## 1. Introduction

The basic aim of the application of data fusion is to reduce the uncertainty of measurement results or to increase the effectiveness of classification, detection, and location of the object. This idea involves the joint use of signals and measurement data from many sensors and information derived from other sources (e.g., a priori knowledge). The method used to combine this information depends on the specifics of the object being measured and on the measurement tools used. This method can take diverse forms, starting with a simple averaging of results and continuing to the use of models based on Bayes theory or Kalman’s filtration theory [1,2,3,4,5,6].

The methods used for the fusion of data can be divided into three groups:competitive fusion, where different types of sensors are used to measure the same physical quantity. This may lead to information redundancycomplementary fusion, where each sensor is used to measure a different property of the studied objectcooperative fusion, where the correct operation of a single sensor is dependent on the results of some other sensor. Without cooperation, the operation of the first sensor would be impossible or undesirable.

One specific measurement system in which data fusion brings significant benefits is the vehicle Multi-Sensor Weigh-In-Motion (MS-WIM) system. The objective of measurement in such a WIM system is the quantitative determination of the vehicle’s static axle loads and its GVW (gross vehicle weight) with higher accuracy than in classical WIM systems [7,8,9,10,11].

The key challenge is to provide WIM (Weigh-in-Motion) systems with high accuracy of weighing, comparable to systems that operate in static conditions (1 ÷ 2% for GVW). Such accuracy ensures the effective use of WIM systems in mass enforcement systems to protect the traffic infrastructure. It is also important to ensure the possibility of maintaining this accuracy over the system’s lifetime. A significant barrier to the realization of these aims results from the sensitivity of WIM systems to factors that produce inconsistent weight measurements (e.g., the pavement condition and its temperature, vehicle speed and the type of its suspension, vibrations caused by a vehicle, environmental factors, etc.) [12,13,14,15,16].

Currently, in many countries, intensive studies are being conducted aimed at solving technical and legislative problems that would allow for the application of WIM systems for direct mass enforcement [17,18,19,20,21,22,23,24,25]. One of the conditions for such application of WIM systems is to ensure high and constant weighing accuracy. The accuracy of vehicle weighing is determined by two components: systematic error (bias) and random variability of results (uncertainty). Bias is minimized as a result of WIM system calibration. In the paper, we present the results of a comparison of two algorithms for measurement results fusion. These measurement results are obtained from load sensors installed in the WIM system. As the comparison criterion, we adopted the uncertainty of the final result, i.e., the gross vehicle weight and the static load of each axle.

In our opinion, the use of data fusion in WIM systems allows for a reduction in the sensitivity of weighing results to a disturbing factor. At the same time, it will increase and stabilize weighing accuracy over a long timeframe. In effect, this allows for an increase in the effectiveness of the protection of the road infrastructure from damage caused by overloaded vehicles.

In this paper, we presented a comparison of two estimators used to estimate the static axle load and the gross vehicle weight. The comparison was based on synthetic data generated by computer simulation and on experimental data. We obtained experimental data from a 16-sensor WIM system. The estimators compared were the mean value and the maximum likelihood estimator (MLE). The results presented in this paper were obtained by applying competitive fusion. Competitive fusion takes into account the uncertainty of measurement of a given load sensor in the system and the influences of this uncertainty on the static load estimate.

The paper is organized in the following manner. In Section 2 and Section 3, estimators of GVW or the static load of an individual axle are presented which enable measurement results obtained from each of the 16 load sensors to be fused. These are respectively a mean value and the maximum likelihood estimator. The properties of these estimators are evaluated based on synthetic measurement data, data which was obtained from simulations conducted. A discussion is presented in Section 4. This section contains also the results of experimental studies conducted at a real WIM site equipped with 16 load sensors. The experiment conducted is described along with the results of a comparison of both estimators used for processing measurement data. The assessment criterion for the estimators was the uncertainty of the estimation of the GVW and static load of individual axles. Finally, Section 5 concludes this paper.

## 2. Mean Value as an Estimator of Static Axle Load and GVW

In MS-WIM systems, each axle is weighed a number of times equal to the number of load sensors installed in the system. If the MS-WIM system contains *Nsens* load sensors, a vehicle with a number of axles equal to *Naxles* generates *Nsens · Naxles* load measurement results. In order to limit the random variability of the weighing results, the final measurement result is the effect of an average of the weighing results from the individual sensors. One frequently applied averaging algorithm is the mean value (1):(1)$$Nc_{mean}=\frac{1}{Nsens}\sum_{i=1}^{Nsens}y_{i}$$
where: $Nc_{mean}$—the sought value, which is the estimate of the static load of a selected axle or the GVW, dependent on the solution adopted, Nsens—the number of load sensors in the MS-WIM system, $y_{i}$—the result of selected axle weighing from the *i*-th sensor or sum of the weighing of all axles of the vehicle from this sensor, dependent on the solution adopted.

The weighing result $y_{i}$ may be understood as the load of a single axle or the sum of all axles on the *i*-th sensor. In the first instance, estimator (1) allows an estimate of static axle load on a given axle to be determined, while in the second instance an estimate of the GVW is made.

## 3. Maximum Likelihood Estimator

Maximum Likelihood (ML) algorithms are used to solve many problems in the field of signal analysis and identification of objects [26]. These allow knowledge of the studied object derived from two sources to be combined, i.e., knowledge from an experimental study of the object (measurement data) and expert knowledge resulting from experience. A condition for the practical application of a maximum likelihood algorithm is knowledge of the probability distribution of measurement results.

The ML method allows for the determination of the coefficients of a model describing the identified object. The determined coefficients maximize the probability of a certain event. This event involves the occurrence of such values of measurement results that were actually obtained during an experiment carried out on this object.

In MS-WIM systems, the value directly measured is the dynamic load of all axles of the weighed vehicle, as assessed by successive load sensors. Expert knowledge in this case involves knowledge of the standard deviation of load measurement results from successive sensors. The source of expert knowledge is the results of previous, periodic calibrations of the WIM system. This knowledge can be obtained by collecting results from individual sensors with repeated passages of vehicles with known axle loads (e.g., during system calibration or by identifying in traffic a special group of reference vehicles) [13,14,15,16,17].

The discrepancies which arise among the sensors results from the lack of full repeatability of manufacture and installation of sensors in the pavement, and are also the result of the eventual degradation of the individual sensors, which occurs at different rates. Moreover, it can be assumed that the measurement results from individual sensors are statistically independent and that the distribution of the probability of these results is known. Taking into consideration the above assumptions, the measurement model may be formulated as (2):(2)$$y_{i}=Nc+\varepsilon_{i}$$
where: Nc—the sought value, which dependent on the solution adopted, is the static load of a selected axle or the GVW, established as the sum of the static load of all axles, $\varepsilon_{i}$—the random additive component, i=1, 2, 3, …Nsens—the number of the load sensor.

Assuming a normal distribution of the random component $\varepsilon_{i}$, its expected value equal to zero, and the statistical independence of these components for individual load sensors, the joined probability density p(ε) can be described by (3):(3)$$p\left(\varepsilon \right)=\overset{Nsens}{\underset{i=1}{\prod }}\left(\frac{1}{\sqrt{2\pi }\sigma_{i}}\;e^{-\frac{\varepsilon_{i}}{2\sigma_{i}^{2}}}\right)$$
where $\sigma_{i}$ is the variable standard deviation of $\varepsilon_{i}$ for *i*-th sensor.

Based on Equation (2), $\varepsilon_{i}=y_{i}-Nc$. Thus, the joined probability density of the measurement results for all the load sensors is described by (4). This is known as the likelihood function.
(4)$$p\left(y_{1},\;y_{2},\ldots y_{Nsens}/Nc\right)=\overset{Nsens}{\underset{i=1}{\prod }}\left(\frac{1}{\sqrt{2\pi }\sigma_{i}}\;e^{-\frac{{\left(y_{i}-Nc\right)}^{2}}{2\sigma_{i}^{2}}}\right)={\left(\frac{1}{\sqrt{2\pi }\sigma_{i}}\right)}^{Nsens}e^{-\sum_{i=1}^{Nsens}\frac{{\left(y_{i}-Nc\right)}^{2}}{2\sigma_{i}^{2}}\;}$$

The key idea of an ML estimator is a search is made for a value for the sought quantity Nc, for which the likelihood function (4) reaches its maximum value.

The problem of searching for the maximum likelihood function (4) can for simplicity’s sake be substituted with an equivalent problem of searching for the maximum of the logarithmic likelihood function (5):(5)$$ln\left[p\left(y_{1},\;y_{2},\ldots y_{Nsens}/Nc\right)\right]=C-\overset{Nsens}{\underset{i=1}{\sum }}\frac{{\left(y_{i}-Nc\right)}^{2}}{2\sigma_{i}^{2}}$$
where $C={\left(\frac{1}{\sqrt{2\pi }\sigma_{i}}\right)}^{Nsens}$ is a constant not dependent on the measured quantity Nc.

Achievement of the maximum by function (5) means that condition (6) has been satisfied.
(6)$$\frac{\partial ln\left[p\left(y/Nc\right)\right]}{\partial Nc}=-\overset{Nsens}{\underset{i=1}{\sum }}\frac{\left(y_{i}-Nc\right)}{2\sigma_{i}^{2}}\left(-2\right)=\overset{Nsens}{\underset{i=1}{\sum }}\frac{y_{i}-Nc}{\sigma_{i}^{2}}=0$$

The solution of Equation (6) leads to a maximum likelihood estimator (MLE) for the desired value Nc:(7)$$Nc_{MLE}=\frac{\sum_{i=1}^{Nsens}\frac{y_{i}}{\sigma_{i}^{2}}}{\sum_{i=1}^{Nsens}\frac{1}{\sigma_{i}^{2}}}$$

If the uncertainty of measurement for all load sensors is uniform, then Equation (7) reduces itself to the mean value (1) of the results from individual sensors. Otherwise, the measurement results obtained from each load sensor are “weighted” depending on their uncertainty.

The ML estimator possesses known properties. It is an asymptotic estimator, i.e., it is merely asymptotically unbiased and asymptotically effective. These features may constitute a significant limitation on its application in weighing results from MS-WIM systems. This is because WIM systems containing many dozens of sensors are rarely used in practice. Generally, such systems are equipped with a few or perhaps a dozen sensors [27]. However, as the results of the study presented in the following part show, the application of estimator (7) is justified even when the MS-WIM system is equipped with 8–10 load sensors.

## 4. Results

This paper presents both the effects of simulation tests and the effects based on the results of measurements carried out at the MS-WIM field site.

### 4.1. Simulation Results

For reasons related to the technical parameters of a specific load sensor, small discrepancies in the installation process of these sensors, and the varying rates of their degradation, successive sensors may vary in terms of the uncertainty. The measure of such uncertainty is understood to be a standard deviation of a sufficiently numerous population of measurement results obtained for the same axle, in repeatable measurement conditions. The experience of the authors suggests that after a two- or three-year period of operation of an MS-WIM system, the standard deviation for the best and worst polymer sensors may differ two- or even threefold. The metrological properties of various load sensors (capacitive, piezoelectric, fiber optic, strain gauges, quartz, etc.) used in WIM systems are presented in detail in the papers [28,29,30,31,32]. In Figure 1a, a standard deviation of measurement results obtained from an exemplary MS-WIM system equipped with 16 piezoelectric load sensors is presented. During the experiments carried out, three vehicles with known mass and static load on each axle repeatedly passed through the tested WIM station. This experiment is described in Section 4.2. Statistical analysis of the weighing results collected in this way allowed the determination of the standard deviation of the results from each load sensor. The standard deviation is expressed as a relative value by reference to the mean value of the weighing results of each axle. Averaging was carried out in the set of weighing results obtained during all test runs, separately for each axle. The measurement uncertainty of the sensors indicated by numbers 3 and 4 clearly deviates from the others.

The properties of Equations (1) and (7) were initially evaluated on the basis of simulation tests. The basis of these studies was synthetic data obtained using a pseudorandom number generator. A normal distribution generator with zero expected value and standard deviation σ was used. The standard deviation for subsequent load sensors was determined on the basis of measurement tests performed on an exemplary MS-WIM system (Figure 1a).

The results of Equation (1) are shown in Figure 1. Two methods were used to average the results from individual sensors. In the first method, the averaging was performed in a natural order resulting from the installation of sensors at the WIM site. In the second case, the averaging was carried out in an order consistent with the growing uncertainty (standard deviation) of the individual sensors. Equation (1) is an unbiased estimator regardless of the order of averaging of the measurement results. If the uncertainty of measurement of all the sensors is uniform, the order of averaging, of course, is not significant, and the larger the number of results which are averaged, the lower is the variability of the estimates calculated based on Equation (1). In this instance, the idea that “the more, the better” applies (curve 4 in Figure 1b). The characteristics for averaging by rising uncertainty of measurement results (curve 3) shows a definite minimum (0.0238), occurring for nine sensors. This result is twice as good as the best result obtained for averaging based on the natural order of the sensors.

The results for Equation (7) are presented in Figure 2. This presents the relative (with reference to the mean value) standard deviation of Equation (7) dependent on the number of load sensors. These characteristics were determined for sensors for the same synthetic data as for Equation (1). Characteristic 1 shows the uncertainty of the estimator when the load measurement results were averaged in order of increasing uncertainty (standard deviation) of these results. For comparison, analogous characteristics are presented for estimator (1)—curve 2.

In the second case, the sensors were arranged in a natural order derived from the order in which they are installed at the MS-WIM site. In this case, the uncertainty of Equation (7) is described by characteristic 3.

From the characteristics presented in Figure 1 and Figure 2, it can be seen that Equation (7) delivers a lower uncertainty of estimate when compared to Equation (1). It thus appears justified to also average measurement results in order of increasing uncertainty of individual sensors and to reject the most uncertain results. Both of these theses were verified experimentally in the next section.

### 4.2. Experimental Results

Experimental studies were conducted on a constructed real site of an MS-WIM system. The site is equipped with 16 polymer piezoelectric load sensors, eight inductive loop sensors, and eight temperature sensors. The load sensors are evenly distributed along the WIM site with 1 m spacing [7,27].

The advantages of MS-WIM systems result first of all from the more efficient averaging of the weighing results of each axle since this is performed over the set of 16 samples of load signal. As a consequence, the disturbing effect caused among others by vertical bouncing of the weighed vehicle and arising from the environment and load sensors themselves (e.g., sensitivity non-uniformity) is more effectively averaged [17].

The measurement experiment comprised multiple weighings of three reference 5-axle vehicles with a known GVW and known static axle load for each individual axle (see this process in Figure 3).

The GVW and axle load were determined with the use of a platform scale, low-speed scale, and devices used for measuring the static load of individual axles. The weighing was conducted before the vehicles passed the MS-WIM station as well as afterward for verification purposes.

Parameters of reference vehicles used are presented in Table 1.

For each run of the reference vehicle, the set of data includes the weighing results for each axle at each of the 16 load sensors. In total, 147 runs were completed giving 16 × 5 × 147 = 11,760 weighing results for individual axles. Because the runs were conducted at different speeds and in changing temperature conditions, it was necessary to introduce corrections to eliminate the impact of these two factors on the weighing result. The correction consisted of removing the trend from the measurement results obtained in subsequent passings of reference vehicles. As a result, data were obtained which were free of trends (Figure 4).

The purpose of the study was to compare Equations (1) and (7) used to estimate the GVW and static load of each axle. A standard deviation was determined for the estimates of GVW and axle load respectively as a criterion for accuracy assessment. As in the case of synthetic data, the weighing results from individual sensors were averaged either in natural sequence resulting from the position of these sensors at the MS-WIM site or in order of increasing standard deviation characterizing their uncertainty. Also, in this case, the uncertainty (standard deviation) of the results from individual sensors was determined as a result of tests carried out on an exemplary MS-WIM system.

In Figure 5 and Figure 6, the dependency between the standard deviation (uncertainty) of the estimate of GVW as a function of the number of sensors is presented. Equations (1) and (7) are compared.

Analogous studies were conducted for the estimates of the static load of individual axles determined by using both compared estimators and results are presented in Figure 7 and Figure 8.

Three five-axle vehicles were used in the experiment as reference vehicles. As examples, the obtained estimation results for two axles, i.e., the first and the fifth, are presented. The first one was chosen as the characteristic axle in this type of vehicle due to the small variability of its load as a function of the total load of the vehicle. The fifth axle is also interesting due to the fact that it is part of the axle group (third, fourth, and fifth axle).

## 5. Discussion

The simulation and experimental results for two vehicles’ weighing results estimation algorithms used at an MS-WIM station are presented in this paper. Both algorithms compared involve the averaging of weighing results from individual load sensors. This averaging may be performed in a natural order resulting from the order in which the sensors are installed at the weighing site. It may also be performed in an order resulting from the accuracy of the sensors, starting from the most accurate sensor. Based on the results presented, the following conclusions may be formulated:MLE Equation (7) allows an estimate of GVW to be obtained with an uncertainty that is no greater than the uncertainty of estimate determined using Equation (1),the benefit derived from the use of Equation (7) depends on the number of sensors installed in the MS-WIM system and on the range of their standard deviations,the properties of Equation (1) depend on the order of averaging of results obtained from successive load sensors,averaging in order from the most accurate sensor to the least accurate allows the uncertainty of results obtained using Equation (1) to be significantly decreased. It also allows the optimum number of sensors to be determined. It is possible to monitor the state of individual load sensors, but this, however, requires further technical operations,MLE Equation (7) is in practical terms not sensitive to the order of averaging. This is due to the fact that an algorithm for eliminating inaccurate sensors is “built-in” to its structure.

## Figures and Tables

**Figure 1 sensors-20-03357-f001:**
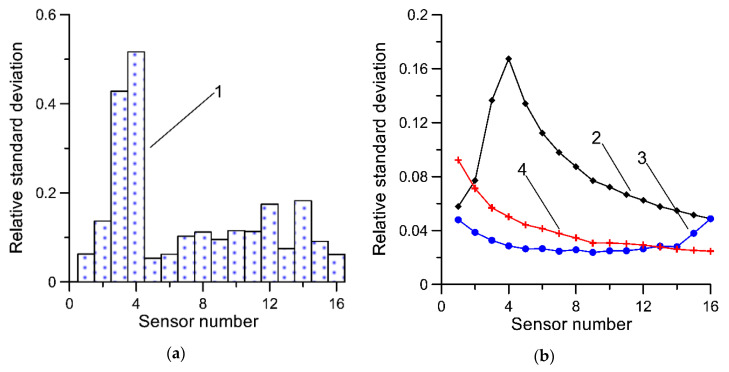
Relative (referring to the average value) standard deviation of estimates calculated on the basis of Equation (1) for various methods of averaging. (**a**) 1—standard deviation arbitrarily assumed for successive load sensors; (**b**) 2—averaging performed based on the order in which the sensors are installed in the pavement, 3—averaging in an order based on the increasing uncertainty of measurement for successive sensors, 4—measurement uncertainty of all sensors is uniform (σ = 0.1). Characteristics determined for synthetic data.

**Figure 2 sensors-20-03357-f002:**
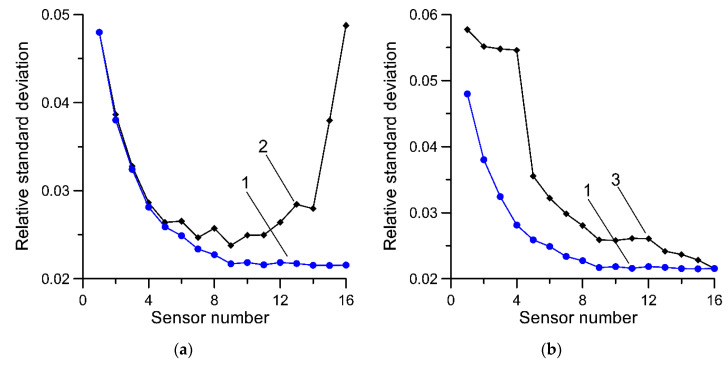
Dependency of relative standard deviation as a function of the number of load sensors for Equation (7). (**a**) 1—the uncertainty of Equation (7) when the load sensors were averaged in order of increasing uncertainty, 2—as a comparison for Equation (1); (**b**) 1—the uncertainty of Equation (7) when the load sensors were averaged in order of increasing uncertainty, 3—the uncertainty of Equation (7) when sensors were arranged in the order in which they are installed at the MS-WIM site.

**Figure 3 sensors-20-03357-f003:**
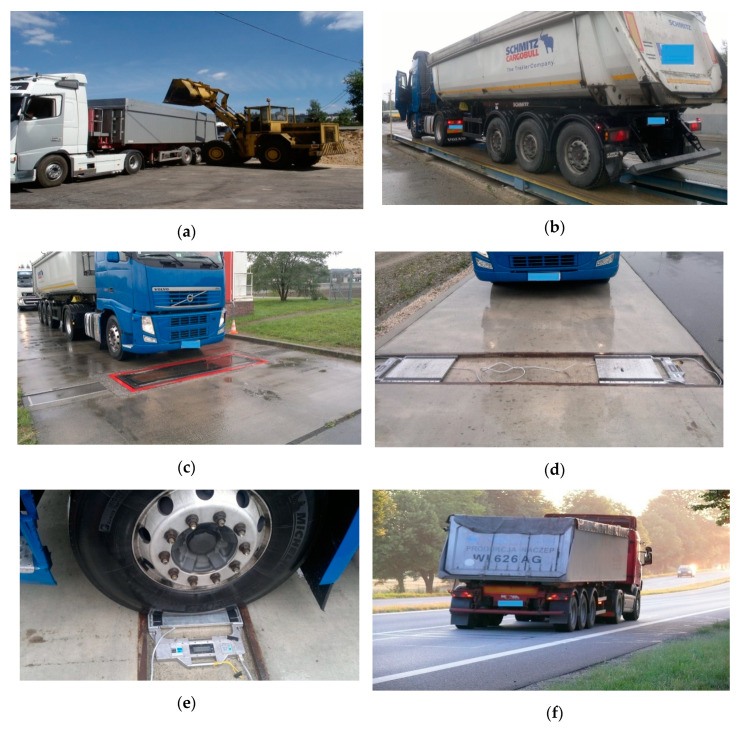
Preparation of reference vehicles: (**a**)—loading the vehicle with gravel; (**b**)—the measurement of GVW on a platform scale; (**c**)—the measurement of axle load and GVW on a low-speed scale; (**d**,**e**)—the measurement of static axle load on a static scale; (**f**)—weighing of the vehicle at the MS-WIM site.

**Figure 4 sensors-20-03357-f004:**
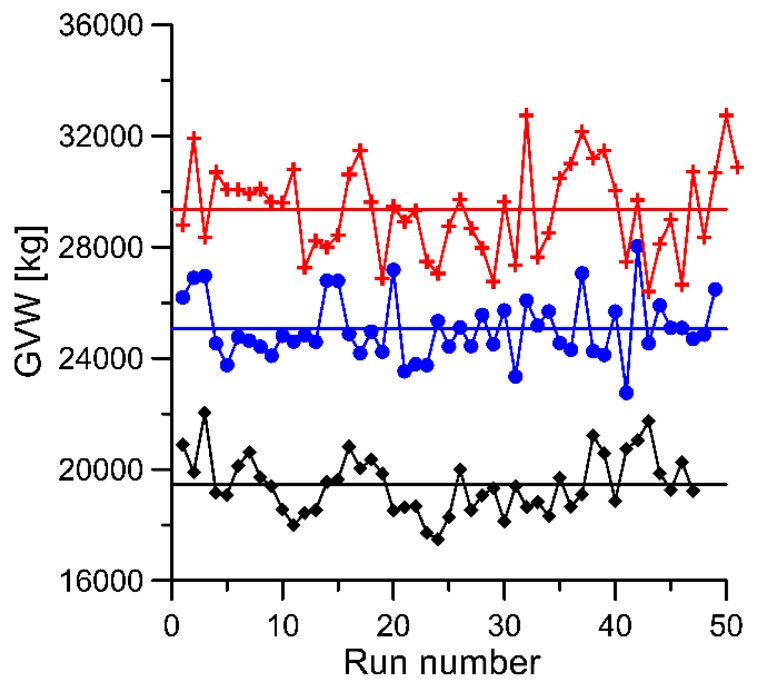
GVW for three reference vehicles in successive runs (as a sum of axle load from the WIM system). black—vehicle no. 1; blue—vehicle no. 2; red—vehicle no. 3 (see Table 1).

**Figure 5 sensors-20-03357-f005:**
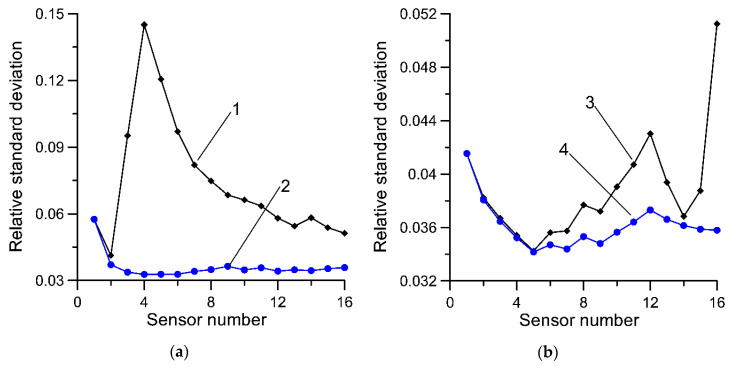
Dependency of the relative standard deviation of results for GVW as a function of the number of load sensors. (**a**)—weighing results for successive load sensors averaged in the order in which the sensors were installed, (**b**—weighing results averaged in order of increasing uncertainty of the results. 1, 3—estimator (1), 2, 4—estimator (7).

**Figure 6 sensors-20-03357-f006:**
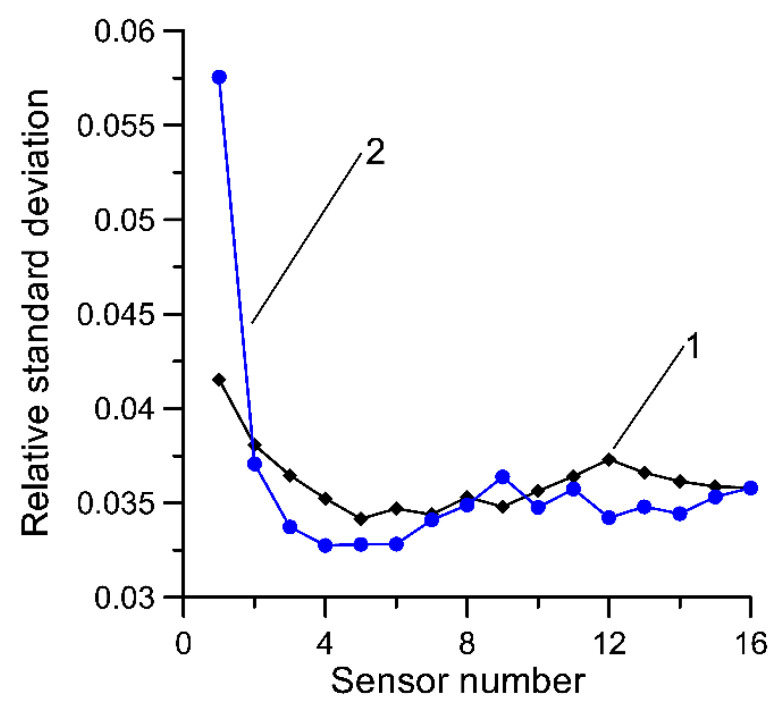
Dependency of the relative standard deviation of weighing results for GVW obtained for Equation (7) as a function of the number of load sensors. 1—weighing results averaged in order of increasing uncertainty of the results, 2—weighing results of successive load sensors averaged in the order in which the sensors were installed.

**Figure 7 sensors-20-03357-f007:**
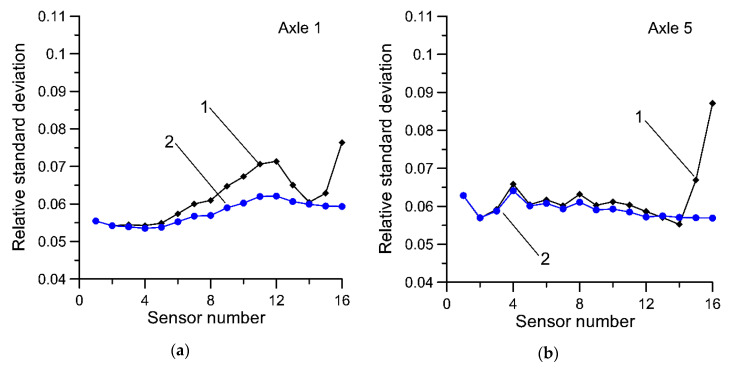
Relative standard deviation of estimates of static axle load, (**a**)—axle no. 1; (**b**)—axle no. 5; with results averaged in order of increasing uncertainty of the results. Curve 1—results for Equation (1), curve 2—results for Equation (7).

**Figure 8 sensors-20-03357-f008:**
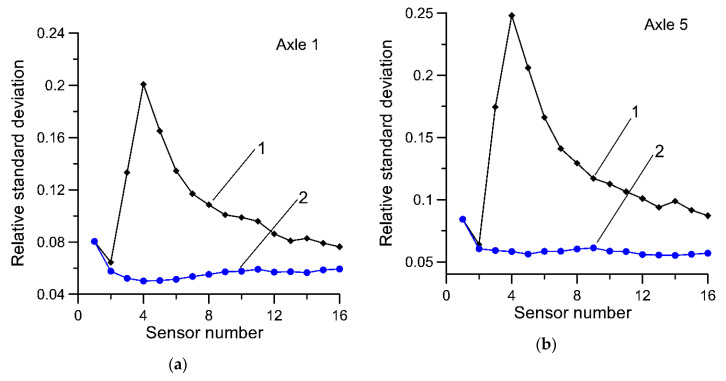
Relative standard deviation of estimates of static axle load, (**a**)—axle no. 1; (**b**)—axle no. 5; with weighing results from individual sensors averaged in the order in which the sensors were installed at the MS-WIM site. Curve 1—results for Equation (1), curve 2—results for Equation (7).

**Table 1 sensors-20-03357-t001:** Parameters of reference vehicles.

Vehicle No.	GVW [kg]	Axle Load [kg]					No. of Runs
		1	2	3	4	5	
1	19,460	6044	5505	2604	2604	2703	47
2	25,060	6645	7229	3526	3805	3855	49
3	29,360	6436	7328	5149	5248	5199	51
Total							147

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
