# Peer review of "Sensor Data Fusion in Multi-Sensor Weigh-In-Motion Systems"

_sensors, 2020, doi:10.3390/s20123357_

Round 1

Reviewer 1 Report

    Sensor Data Fusion in Multi Sensor WIM systems   - Abstract: Poorly written.  Does not convey purpose of the work and doesn't provide sufficient detail about the "estimators" to appreciate the methodology presented in the paper. Overly vague.   -Methodology: Method minimizes the relative standard deviation of sensor measurements.  Can the authors explain how their work addresses the issue that standard deviation can be minimized (high precision) but the measurements could be incorrect (low accuracy)? Consider a scenario where the sensor array was mis-calibrated at installation.  It is not clear how the static weights of the test vehicles in the experiment were used in the assessment. There should be a table showing weighing errors for each test run given the two averaging methods.  Perhaps this is what the authors have embedded in the relative standard deviation, but it is not clear.  Thus a formulation for calculating relative standard deviation in this context would be helpful in understanding the averaging and MLE methods and the ordering of sensor measurements within those averages.   The paper lacks a sensitivity analysis to speed and weight of the test trucks and the degree of mis-calibration of the sensor array and individual sensors.  Without a sensitivity analysis, one cannot truly judge the performance of the two estimators.     Specific issues: 1. Define WIM and GVW as acronyms around line 40 2. What is the meaning of 1-2% GVW on line 42? 3. Applications of WIM need to be mentioned near line 40 to appreciate the context of the problem. e.g., enforcement of truck loading to protect pavement and bridge infrastructure.  This  is later mentioned but would have value mentioning sooner in the paper.  4. The phrase "disturbing" factors on line 44 can be better phrased as "Factors that produce inconsistent weight measurements" 5. The paper needs light grammar editing.  e.g., Line 94, "The ML consists of..." 6. The authors point out that measurement errors are a result of installation and degradation (line 143).  They should also note that discrepancies are the result of vehicle dynamics over the sensor.  This is a concern especially for large trucks with liquid bulk content that shifts position and thus axle weight over the sensor.  7. The authors mention polymer sensor accuracy (line 149).  I think a more specific weight measurement accuracy profile by sensor type would benefit the relevance of the work. More specific details on measurement error ranges is needed in general, e.g., line 149, two or three fold is too vague. 8. I am unclear of what the graphs in Figures 1 and 2 represent.  If the horizontal axis is meant to show the "number of sensors" then I follow the logic in the statements beginning on line 158.  If it is "sensor number" in reference to the position of the sensor in the array, then I do not understand what the Figures convey or how they were determined. For the case where sensors are averaged in order of increasing uncertainty, I am unclear if the horizontal axis in the figure means the number of sensors or the order of sensors.  In this case does the order change for each alternative (1), (2), etc.? In figure 2, I am unsure what the black line labeled as (2) represents in (a), e.g., is it the same order of sensors but using the averaging estimator? 9. A schematic to accompany the photos of the sensor array would be helpful to understand if the sensors are an array in the direction of travel or in a direction perpendicular to the direction of travel.   10. Figure 3a and b and e and f are not necessary. 11. Explanation of the correction procedures is needed (line 217).  12. The rationale for the selection of vehicle weights and speeds is needed.  Were they randomly selected or was there an underlying hypothesis about how the ML and averaging estimators may perform given varied weight and speed profiles of test trucks? 13.  Referencing the first and fifth axles is beneficial for a sensitivity analysis. However, the statement on line 244 about the axle group is unclear.  Do the authors mean to say that the fifth axle supports the load and therefore has more variability?  

Reviewer 2 Report

Overview

The authors present results of a study comparing two statistical estimators for determining the axle loads and gross vehicle weight using weigh-in-motion systems. The first estimator is a simple mean value estimator, while the second one is the maximum likelihood estimator derived based on the assumption of statistical independence of sensor measurements and normal probability density distribution of measurement uncertainty. The paper presents results of numerical investigation supported by the results obtained from experimental measurements from a real weigh-in-motion system with 16 sensors. The authors conclude that the maximum likelihood estimator is able to reduce uncertainty in the estimated axle loads and gross vehicle weight using a relatively small number of sensors.

Overall, the paper is well-written. The conclusions made by authors are supported by the data presented in the paper. The reviewer recommends the paper for publication after the authors address several minor remarks listed below.  

Remarks:

  1. Page 2, lines 59-60, please rephrase the sentence: “The competitive fusion taking into account the uncertainty of measurement of a given load sensor in the 60 system and its influences on the estimation of static load.”
  2. Lines 156-157 – please clarify if the standard deviation used in simulation was obtained from experimental study. It seems like it does, but still is unclear.
  3. Please define using an equation “relative standard deviation” in Fig. 1 and others.
  4. Line 149: “may be differ” should be “ may differ”.
  5. Line 151: “clear” should be “clearly”.
  6. Line 152: “from the others” should be “from others”.
